# On the separation of Hall and Ohmic nonlinear responses

Stepan S. Tsirkin[1⋆] and Ivo Souza[2,3]

**1** Physik-Institut, Universität Zürich, Winterthurerstrasse 190, CH-8057 Zürich, Switzerland
**2** Centro de Física de Materiales, Universidad del País Vasco, 20018 San Sebastián, Spain
**3** Ikerbasque Foundation, 48013 Bilbao, Spain

⋆ stepan@physik.uzh.ch

## Abstract

The symmetric and antisymmetric parts of the linear conductivity describe the dissipative (Ohmic) and nondissipative (Hall) parts of the current. The Hall current is always transverse to the applied electric field regardless of its orientation; the Ohmic current is purely longitudinal in cubic crystals, but in lower-symmetry crystals it has a transverse component whenever the field is not aligned with a principal axis. In this work, we extend that analysis beyond the linear regime. We consider all possible ways of partitioning the current at any order in the electric field without taking symmetry into account, and find that the Hall vs Ohmic decomposition is the only one that satisfies certain basic requirements. A general prescription is given for achieving that decomposition, and the case of the quadratic conductivity is analyzed in detail. By performing a symmetry analysis we find that in five of the 122 magnetic point groups the quadratic dc conductivity is purely Ohmic and even under time reversal, a type of response that is entirely disorder mediated.

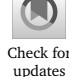

# 1 Introduction

A static electric field applied to a conducting crystal generates a current density that may be written to linear order as

$$j_\alpha^{(1)} = \sigma_{\alpha\beta} E_\beta \,, \tag{1}$$

where a summation over Cartesian index $\beta$ is implied, and $\sigma_{\alpha\beta}$ is understood to be a function of the externally applied magnetic field $\boldsymbol{H}$. In general $\boldsymbol{j}^{(1)}$ is not parallel to $\boldsymbol{E}$, but under certain conditions it may contain a part that is always perpendicular to $\boldsymbol{E}$, irrespective of how the field is oriented relative to the crystal axes. This *Hall current* is described by the antisymmetric part of the linear conductivity tensor,

$$j_{\mathcal{H},\alpha}^{(1)} = \sigma_{\alpha\beta}^{\mathcal{H}} E_\beta \,, \qquad \sigma_{\alpha\beta}^{\mathcal{H}} = \frac{1}{2}\big(\sigma_{\alpha\beta} - \sigma_{\beta\alpha}\big), \tag{2}$$

and the remainder $\boldsymbol{j}_{\mathcal{O}}^{(1)} = \boldsymbol{j}^{(1)} - \boldsymbol{j}_{\mathcal{H}}^{(1)}$, given by the symmetric part of the conductivity, is the *Ohmic current* that gives rise to energy dissipation via Joule heating,

$$j_{\mathcal{O},\alpha}^{(1)} = \sigma_{\alpha\beta}^{\mathcal{O}} E_\beta \,, \qquad \sigma_{\alpha\beta}^{\mathcal{O}} = \frac{1}{2}\big(\sigma_{\alpha\beta} + \sigma_{\beta\alpha}\big). \tag{3}$$

Building on seminal works from the 1970s [1–3], there is at present renewed interest in *nonlinear* effects in solids arising from broken symmetries [4]. The nonlinear transport effects that are being actively investigated include unidirectional magnetoresistance (both induced by a magnetic field [5–7] and spontaneous [8–10]), and various nonlinear Hall effects [11–24].

Despite the surge of interest in nonlinear currents, a clear discussion of how to extend the Hall vs Ohmic decomposition to the nonlinear regime is lacking, and confusing or even incorrect statements can be found in the recent literature. With the present work we aim to clarify the phenomenology of the nonlinear Hall vs Ohmic decomposition, and to place it in the broader context of how to partition the nonlinear current into physically well-defined parts. Although we will focus on the conductivity tensor, our analysis applies equally well to the resistivity. For simplicity, we will assume throughout that the applied electric field is static.

To motivate the problem, consider the second-order response

$$j_\alpha^{(2)} = \sigma_{\alpha\beta\gamma} E_\beta E_\gamma \,, \tag{4}$$

which requires broken inversion symmetry. Contrary to the linear conductivity, the quadratic conductivity is not uniquely defined since adding to it a correction of the form

$$\Delta\sigma_{\alpha\beta\gamma} = -\Delta\sigma_{\alpha\gamma\beta} \tag{5}$$

does not change the physically observable current. We will refer to this freedom in defining nonlinear conductivities as a "gauge freedom," and to the unique choice that satisfies $\sigma_{\alpha\beta\gamma} = \sigma_{\alpha\gamma\beta}$ as the "symmetric gauge." Thus, the symmetric gauge is the one where the conductivity tensor has intrinsic permutation symmetry [25].

By analogy with Eq. (2), one might attempt to define $\sigma_{\alpha\beta\gamma}^{\mathcal{H}}$ as the part of $\sigma_{\alpha\beta\gamma}$ that is antisymmetric in either the first and second indices,

$$\sigma_{\alpha\beta\gamma}^{1,2} = \frac{1}{2}\big(\sigma_{\alpha\beta\gamma} - \sigma_{\beta\alpha\gamma}\big) \,, \tag{6}$$

or in the first and third,

$$\sigma_{\alpha\beta\gamma}^{1,3} = \frac{1}{2}\big(\sigma_{\alpha\beta\gamma} - \sigma_{\gamma\beta\alpha}\big) \,. \tag{7}$$

(Note that we need to choose between these two options, since imposing both conditions would render $\sigma^{\mathcal{H}}_{\alpha\beta\gamma}$ totally antisymmetric, resulting in zero current.) Both choices yield Hall-like transverse currents. However, not only do they give different currents, but those currents depend on the initial gauge choice for $\sigma_{\alpha\beta\gamma}$. Both problems can be fixed by switching to the symmetric gauge, $\overline{\sigma}_{\alpha\beta\gamma} = \frac{1}{2}\left(\sigma_{\alpha\beta\gamma} + \sigma_{\alpha\gamma\beta}\right)$, before applying the antisymmetrization (6) or (7). Since the resulting Hall-like conductivities

$$\overline{\sigma}^{1,2}_{\alpha\beta\gamma} = \frac{1}{4}\left(\sigma_{\alpha\beta\gamma} + \sigma_{\alpha\gamma\beta} - \sigma_{\beta\alpha\gamma} - \sigma_{\beta\gamma\alpha}\right), \tag{8}$$

and

$$\overline{\sigma}^{1,3}_{\alpha\beta\gamma} = \frac{1}{4}\left(\sigma_{\alpha\beta\gamma} + \sigma_{\alpha\gamma\beta} - \sigma_{\gamma\beta\alpha} - \sigma_{\gamma\alpha\beta}\right) \tag{9}$$

satisfy $\overline{\sigma}^{1,3}_{\alpha\beta\gamma} = \overline{\sigma}^{1,2}_{\alpha\gamma\beta}$, they clearly yield the same current (they are related by the gauge transformation $\Delta\sigma_{\alpha\beta\gamma} = \overline{\sigma}^{1,2}_{\alpha\gamma\beta} - \overline{\sigma}^{1,2}_{\alpha\beta\gamma}$). This modified prescription [15,19] is nevertheless still not quite correct.

As a concrete example, we take the expression for the quadratic conductivity obtained by solving the Boltzmann equation at $H = 0$ in the constant relaxation-time approximation. Denoting the relaxation time as $\tau$, there are contributions of order $\tau^0$, $\tau^1$, and $\tau^2$, with those of even (odd) order in $\tau$ being odd (even) under time reversal $\mathcal{T}$ [26] . Neglecting disorder-mediated contributions (skew-scattering and side jump) one finds [10,11,13,14,21]

$$\sigma_{\alpha\beta\gamma} = \frac{e^3}{\hbar}\int_{\boldsymbol{k}n} f_0(\epsilon_n)\left[\left(\partial_\alpha G^{\beta\gamma}_n - \partial_\beta G^{\alpha\gamma}_n\right) + (\tau/\hbar)\partial_\gamma\Omega^{\alpha\beta}_n - (\tau/\hbar)^2\partial^3_{\alpha\beta\gamma}\epsilon_n\right], \tag{10}$$

where $\int_{\boldsymbol{k}n} \equiv \int d^dk/(2\pi)^d\sum_n$ in $d$ dimensions and we have dropped $\boldsymbol{k}$ from the integrand, $e > 0$ is the elementary charge, $\epsilon_n$ is the band energy, $f_0$ is the Fermi-Dirac distribution function, and $\partial_\gamma \equiv \partial/\partial k_\gamma$. $\Omega^{\alpha\beta}_n$ is the Berry curvature, and $G^{\alpha\beta}_n$ is sometimes called the Berry curvature polarizability; these two quantities can be expressed in terms of the Berry connection matrix $A^\alpha_{mn} = i\langle u_m|\partial_\alpha u_n\rangle$ as follows,

$$\Omega^{\alpha\beta}_n = \partial_\alpha A^\beta_{nn} - \partial_\beta A^\alpha_{nn} = -2\mathrm{Im}\langle\partial_\alpha u_n|\partial_\beta u_n\rangle, \tag{11}$$

$$G^{\alpha\beta}_n = -2\mathrm{Re}\sum_m^{\epsilon_m\neq\epsilon_n}\frac{A^\alpha_{nm}A^\beta_{mn}}{\epsilon_n - \epsilon_m}. \tag{12}$$

The $\mathcal{O}(\tau^0)$ and $\mathcal{O}(\tau^1)$ terms in in Eq. (10) describe respectively $\mathcal{T}$-odd and $\mathcal{T}$-even quadratic anomalous Hall responses whose net current we denote by $\boldsymbol{j}^{(2)}_{\mathcal{H}}$, and the $\mathcal{O}(\tau^2)$ term is a $\mathcal{T}$-odd Drude-like quadratic conductivity that has been identified as a mechanism for spontaneous unidirectional magnetoresistance [10]. Applying to Eq. (10) each of the prescriptions in Eqs. (6-9), we obtain

$$\left(\boldsymbol{j}^{1,2}, \boldsymbol{j}^{1,3}, \overline{\boldsymbol{j}}^{1,2} = \overline{\boldsymbol{j}}^{1,3}\right) = (1, {}^1\!/_2, {}^3\!/_4)\,\boldsymbol{j}^{(2)}_{\mathcal{H}} \tag{13}$$

for the quadratic Hall currents. Prescription (6) gives the full Hall current $\boldsymbol{j}^{(2)}_{\mathcal{H}}$, but that is accidental: if we make the gauge transformation $\sigma_{\alpha\beta\gamma} \to \sigma_{\alpha\gamma\beta}$ in Eq. (10), the Hall currents obtained from prescriptions (6) and (7) get swapped: $\left(\boldsymbol{j}^{1,2}, \boldsymbol{j}^{1,3}\right) \to ({}^1\!/_2, 1)\,\boldsymbol{j}^{(2)}_{\mathcal{H}}$. We mentioned earlier that the prescriptions in Eqs. (8) and (9) are not quite correct, and indeed they only recover three quarters of the full Hall current; we will see in Sec. 3 that multiplying the right-hand sides of those equations by factors of 4/3 does lead to generally valid expressions for the quadratic Hall conductivity.

The strategies in Eqs. (6-9), which constitute attempts to generalize to third-rank tensors the definition in Eq. (2) of an antisymmetric tensor of rank two, fail to yield a proper decomposition of the quadratic current. On the other hand, higher-order generalizations of the symmetrization procedure in Eq. (3) are straightforward, since one can symmetrize over all indices. In the case of the quadratic conductivity one finds

$$\sigma^{\mathcal{O}}_{\alpha\beta\gamma} = \frac{1}{6}\left(\sigma_{\alpha\beta\gamma} + \sigma_{\alpha\gamma\beta} + \sigma_{\beta\alpha\gamma} + \sigma_{\beta\gamma\alpha} + \sigma_{\gamma\alpha\beta} + \sigma_{\gamma\beta\alpha}\right), \qquad (14)$$

and it can be readily checked that the power dissipation is fully accounted for by $\sigma^{\mathcal{O}}_{\alpha\beta\gamma}$,

$$\boldsymbol{j}^{(2)} \cdot \boldsymbol{E} = \sigma_{\alpha\beta\gamma} E_\alpha E_\beta E_\gamma = \sigma^{\mathcal{O}}_{\alpha\beta\gamma} E_\alpha E_\beta E_\gamma, \qquad (15)$$

which justifies calling it the quadratic Ohmic conductivity. Accordingly,

$$\sigma^{\mathcal{H}}_{\alpha\beta\gamma} = \sigma_{\alpha\beta\gamma} - \sigma^{\mathcal{O}}_{\alpha\beta\gamma} \qquad (16)$$

describes the dissipationless (Hall) part of the quadratic current response.

Surprisingly we could not find, in the growing literature on nonlinear currents in solids, any explicit mention of the simple prescription in Eqs. (14) and (16) for separating the nonlinear Hall and Ohmic conductivities. Let us apply it to the expression in Eq. (10) for the quadratic conductivity. Since the $\mathcal{O}(\tau^0)$ and $\mathcal{O}(\tau^1)$ terms therein are antisymmetric in two indices, they drops out from Eq. (14); and since the $\mathcal{O}(\tau^2)$ is already totally symmetric, it becomes the full $\sigma^{\mathcal{O}}_{\alpha\beta\gamma}$. Hence, the former terms are Hall-like and the latter is Ohmic.

It should be noted that we have not yet proven that Eqs. (14) and (16) give the *only* valid decomposition of the quadratic current into Ohmic and Hall parts. For example, one could define another partition

$$\tilde{\sigma}^{\mathcal{H}}_{\alpha\beta\gamma} = (1-x)\sigma^{\mathcal{H}}_{\alpha\beta\gamma}, \qquad \tilde{\sigma}^{\mathcal{O}}_{\alpha\beta\gamma} = \sigma^{\mathcal{O}}_{\alpha\beta\gamma} + x\sigma^{\mathcal{H}}_{\alpha\beta\gamma} \qquad (x \in \mathbb{R}) \qquad (17)$$

that is not related to that of Eqs. (14) and (16) by any gauge transformation (5), and again $\tilde{\sigma}^{\mathcal{H}}_{\alpha\beta\gamma}$ would describe a dissipationless current, with all the Joule heating coming from $\tilde{\sigma}^{\mathcal{O}}_{\alpha\beta\gamma}$.

In this work, we consider the problem of defining nonlinear Hall and Ohmic conductivities from a more general perspective. Our starting point is the following question:

*What are all the possible ways of partitioning the nonlinear current into physically meaningful parts, without taking into account neither the symmetries of the system nor specific microscopic mechanisms?*

(We will refer to such partitions as "generic.") To address this question, we start by formulating in Sec. 2 the necessary criteria for a proper generic partition of the current at arbitrary order in $\boldsymbol{E}$. In Sec. 3 we find that there is a *unique* nontrivial decomposition of the current at second order that fulfils those criteria, which corresponds precisely to the Hall vs Ohmic decomposition. (Our criteria do not single out any particular gauges for the partial nonlinear conductivities; instead, they take the form of necessary and sufficient conditions satisfied by the partial conductivities in arbitrary gauges.) The Hall vs Ohmic decomposition is generalized to arbitrary order in Sec. 4. In Sec. 5 we return to the quadratic conductivity to carry out a systematic symmetry analysis of its Hall and Ohmic parts, and in Sec. 6 we draw conclusions. In Appendix A we prove that the Hall vs Ohmic partition of the current is the only generic partition possible at every order in $\boldsymbol{E}$, and in Appendix B we repackage the disorder-free quadratic conductivity (10) in the manner described in Sec. 5.

## 2 Criteria for a generic partition of the current

Our strategy for partitioning the nonlinear current will be as follows. We start from a conductivity tensor $\sigma_{\alpha_0 \alpha_1 \dots \alpha_n}$ describing the full $n$-th order response,

$$j_{\alpha_0}^{(n)} = \sigma_{\alpha_0 \alpha_1 \dots \alpha_n} E_{\alpha_1} \dots E_{\alpha_n}, \tag{18}$$

and search for an operator $\hat{P}$ that selects part of this current. We want the operator $\hat{P}$ to act order by order in the electric field; this means that its action on the full conductivity tensor should result in a linear combination of versions of that same tensor with different sets of indices,

$$\left(\hat{P}\sigma\right)_{\alpha_0 \alpha_1 \dots \alpha_n} = \sum_p a_p \sigma_{\alpha_{p(0)} \alpha_{p(1)} \dots \alpha_{p(n)}}. \tag{19}$$

Here the summation is over all possible mappings

$$\{0, 1, \dots, n\} \xrightarrow{p} \{p(0), p(1), \dots, p(n)\}, \tag{20}$$

where $p(n) \in \{0, 1, \dots, n\}$ and $a_p$ are coefficients to be determined. The part of the current selected by $\hat{P}$ can be written symbolically as

$$\left(\hat{P}\boldsymbol{j}^{(n)}\right)_{\alpha_0} = \left(\hat{P}\sigma\right)_{\alpha_0 \alpha_1 \dots \alpha_n} E_{\alpha_1} \dots E_{\alpha_n}. \tag{21}$$

We shall require three properties of $\hat{P}$. The first is that it acts on the current as a projector, so that

$$\hat{P}\boldsymbol{j}^{(n)} = \hat{P}^2 \boldsymbol{j}^{(n)}, \tag{22}$$

the second is that the projected current is invariant under gauge transformations of the full $n$-th order conductivity tensor, that is,

$$\Delta\left(\hat{P}\boldsymbol{j}^{(n)}\right) = 0 \tag{23}$$

whenever $\Delta \boldsymbol{j}^{(n)} = 0$, which in turn holds if and only if $\Delta\sigma_{\alpha_0 \alpha_1 \dots \alpha_n}$ vanishes under symmetrization over the last $n$ indices.

Finally, we require that the projected current transforms as a vector under rotations of the coordinate system, so that $\hat{P}\boldsymbol{j}^{(n)} \cdot \boldsymbol{E}$ remains invariant under such transformations. This is justified by the intention to arrive at a generic prescription that is not tied to any particular crystal symmetry, and not even to a specific number of spatial dimensions. This third constraint will be satisfied if the summation in Eq. (19) is restricted to *permutation* mappings $p$, for which $p(i) \neq p(j)$ whenever $i \neq j$. Conversely, if mappings with $p(i) = p(j)$ for some $i \neq j$ are included, scalar products will not be conserved under rotations.[1] Thus, from here on we shall restrict our attention to permutation mappings, and investigate which operators $\hat{P}$ can satisfy the two conditions expressed by Eqs. (22) and (23).

Before proceeding, we note that if we find some operator $\hat{P}$ that satisfies the conditions listed above, those conditions will also be satisfied by $\hat{P}' = \hat{1} - \hat{P}$. Thus, any nontrivial operator $\hat{P}$ defines a decomposition of the current into two parts (by "nontrivial" we mean an operator such that $\hat{P}\boldsymbol{j} \neq \boldsymbol{0}$ and $\hat{P}\boldsymbol{j} \neq \boldsymbol{j}$). We will start by applying the above criteria to the second-order response, and then we will generalize to higher orders.

---

[1] Take, for example, $\hat{P}_{\mathcal{H}}\sigma_{\alpha\beta} = (\sigma_{\alpha\alpha} + \sigma_{\beta\beta})/2$. For an electric field lying on the $xy$ plane this gives $\hat{P}_{\mathcal{H}}\boldsymbol{j}^{(1)} \cdot \boldsymbol{E} = E_x^2 \sigma_{xx} + E_y^2 \sigma_{yy} + E_x E_y (\sigma_{xx} + \sigma_{yy})$, and the result should be the same in a different coordinate system. However, in a coordinate system that differs by a two-fold rotation about the $y$ axis we obtain $\hat{P}_{\mathcal{H}}\boldsymbol{j}^{(1)} \cdot \boldsymbol{E} = E_x^2 \sigma_{xx} + E_y^2 \sigma_{yy} - E_x E_y (\sigma_{xx} + \sigma_{yy})$, which is a different result.

## 3 Second-order response

Consider an operator $\hat{P}$ acting on the quadratic conductivity according to Eq. (19),

$$\hat{P}\sigma_{\alpha\beta\gamma} = a_0^+ \sigma_{\alpha\beta\gamma} + a_0^- \sigma_{\alpha\gamma\beta} + a_1^+ \sigma_{\gamma\alpha\beta} + a_1^- \sigma_{\beta\alpha\gamma} + a_2^+ \sigma_{\beta\gamma\alpha} + a_2^- \sigma_{\gamma\beta\alpha}, \tag{24}$$

and on the quadratic current according to Eq. (21),

$$\hat{P}j_\alpha^{(2)} = \left(\hat{P}\sigma_{\alpha\beta\gamma}\right)E_\beta E_\gamma = \left(A_0 \sigma_{\alpha\beta\gamma} + A_1 \sigma_{\beta\alpha\gamma} + A_2 \sigma_{\beta\gamma\alpha}\right)E_\beta E_\gamma. \tag{25}$$

Here $A_i = a_i^+ + a_i^-$, and the notation for the coefficients $a_i^\pm$ is as follows: the subscript denotes the position of $\alpha$ in the permutation of the indices $\alpha\beta\gamma$, and the superscript gives the parity of the permutation.

Our claim is that $\hat{P}$ yields a proper generic partition of the current only if it satisfies Eqs. (22) and (23). Let us start with the gauge-invariance condition (23). The projected current (25) remains unchanged under the gauge transformation (5) if and only if

$$(A_1 - A_2)E_\beta E_\gamma \Delta\sigma_{\beta\alpha\gamma} = 0; \tag{26}$$

since this condition must be satisfied for arbitrary $E$, and since we did not set any rules for permutations involving the first index of $\Delta\sigma$, it follows that $A_1 = A_2$. To impose the idempotency condition (22), we first apply Eq. (24) recursively to find

$$\hat{P}^2\sigma_{\alpha\beta\gamma} = \tilde{a}_0^+ \sigma_{\alpha\beta\gamma} + \tilde{a}_0^- \sigma_{\alpha\gamma\beta} + \tilde{a}_1^+ \sigma_{\gamma\alpha\beta} + \tilde{a}_1^- \sigma_{\beta\alpha\gamma} + \tilde{a}_2^+ \sigma_{\beta\gamma\alpha} + \tilde{a}_2^- \sigma_{\gamma\beta\alpha}, \tag{27}$$

where

$$\tilde{a}_0^+ = a_0^+ a_0^+ + a_0^- a_0^- + a_1^- a_1^- + 2a_2^+ a_1^+ + a_2^- a_2^-, \tag{28a}$$

$$\tilde{a}_0^- = 2a_0^+ a_0^- + a_1^+ a_1^- + a_1^+ a_2^- + a_1^- a_2^+ + a_2^+ a_2^-, \tag{28b}$$

$$\tilde{a}_1^+ = 2a_0^+ a_1^+ + a_0^- a_1^- + a_0^- a_2^- + a_1^- a_2^- + a_2^+ a_2^+, \tag{28c}$$

$$\tilde{a}_1^- = 2a_0^+ a_1^- + a_0^- a_1^+ + a_0^- a_2^+ + a_1^+ a_2^- + a_2^+ a_2^-, \tag{28d}$$

$$\tilde{a}_2^+ = 2a_0^+ a_2^+ + a_0^- a_1^- + a_0^- a_2^- + a_1^+ a_1^+ + a_1^- a_2^-, \tag{28e}$$

$$\tilde{a}_2^- = 2a_0^+ a_2^- + a_0^- a_1^+ + a_0^- a_2^+ + a_1^+ a_1^- + a_1^- a_2^+. \tag{28f}$$

By analogy with Eq. (25) we have

$$\hat{P}^2 j_\alpha^{(2)} = \left(\tilde{A}_0 \sigma_{\alpha\beta\gamma} + \tilde{A}_1 \sigma_{\beta\alpha\gamma} + \tilde{A}_2 \sigma_{\beta\gamma\alpha}\right)E_\beta E_\gamma \tag{29}$$

for the twice-projected current, where the coefficients $\tilde{A}_i = \tilde{a}_i^+ + \tilde{a}_i^-$ are given by

$$\tilde{A}_0 = A_0^2 \quad + (a_1^- + a_2^+)A_1 + (a_1^+ + a_2^-)A_2, \tag{30a}$$

$$\tilde{A}_1 = A_0 A_1 + (a_0^+ + a_2^-)A_1 + (a_0^- + a_2^+)A_2, \tag{30b}$$

$$\tilde{A}_2 = A_0 A_2 + (a_0^- + a_1^+)A_1 + (a_0^+ + a_1^-)A_2. \tag{30c}$$

Equating (25) and (29), the idempotency condition becomes $A_i = \tilde{A}_i$ for $i = 0, 1, 2$. Substituting Eq. (30) for $\tilde{A}_i$ and then invoking the gauge invariance condition $A_1 = A_2$, we are left with two conditions only,

$$A_0 = A_0^2 + 2A_1^2, \qquad A_1 = (2A_0 + A_1)A_1. \tag{31}$$

These equations have four solutions. There are two solutions with $A_1 = 0$,

$$\begin{cases} \hat{P}_0 : (A_0, A_1 = A_2) = (0, 0) \\ \hat{P}_1 : (A_0, A_1 = A_2) = (1, 0) \end{cases} \Rightarrow \quad j^{(2)} = 0 + j^{(2)}, \tag{32}$$

which as indicated give the trivial "all or nothing" partition of the current. Then there are two solutions with $A_1 \neq 0$,

$$\begin{cases} \hat{P}_{\mathcal{H}} : (A_0, A_1 = A_2) = (\frac{2}{3}, -\frac{1}{3}) \\ \hat{P}_{\mathcal{O}} : (A_0, A_1 = A_2) = (\frac{1}{3}, \frac{1}{3}) \end{cases} \Rightarrow \boldsymbol{j}^{(2)} = \boldsymbol{j}^{(2)}_{\mathcal{H}} + \boldsymbol{j}^{(2)}_{\mathcal{O}}, \tag{33}$$

which give the desired Hall vs Ohmic partition. To show that this is the case, we turn to the condition that defines a Hall-like projected current,

$$\hat{P}\boldsymbol{j}^{(2)} \cdot \boldsymbol{E} = 0, \quad \forall \boldsymbol{E}. \tag{34}$$

Using Eq. (25) that condition becomes $A_0 + A_1 + A_2 = 0$, which is satisfied by $\hat{P}_{\mathcal{H}}$ but not by $\hat{P}_{\mathcal{O}}$. This conclude the proof that Eqs. (22) and (23) lead to a partition of the quadratic current into Hall and Ohmic parts. Remarkably, we found that this is in fact the only gauge-invariant and idempotent generic partition possible, apart from the trivial one in Eq. (32).

Since we are still free to adjust the six coefficients $a_i^{\pm}$ in Eq. (24) as long as $A_i = a_i^{+} + a_i^{-}$ maintain the values given in Eq. (33), the Hall and Ohmic quadratic conductivities are highly nonunique. This nonuniqueness corresponds precisely to the gauge freedom (5) in defining $\sigma_{\alpha\beta\gamma}^{\mathcal{H}}$ and $\sigma_{\alpha\beta\gamma}^{\mathcal{O}}$, and it does not affect the physical currents $\boldsymbol{j}^{(2)}_{\mathcal{H}}$ and $\boldsymbol{j}^{(2)}_{\mathcal{O}}$. One way to fulfill the "Ohmic" conditions in Eq. (33) is by setting all six coefficients in Eq. (24) to 1/6, which leads to the fully symmetric form for $\sigma_{\alpha\beta\gamma}^{\mathcal{O}}$ in Eq. (14).

Let us now revisit the prescriptions proposed in Eqs. (8) and (9) for defining $\sigma_{\alpha\beta\gamma}^{\mathcal{H}}$, which consist in first symmetrizing the full $\sigma_{\alpha\beta\gamma}$ in the last two indices, and then antisymmetrizing the first index with either the second or the third [15, 19]. When applied to a concrete example in Sec. 2, those prescriptions only recovered three quarters of the full Hall current [see Eq. (13)]. This suggests it may be possible to fix them by multiplying each of Eqs. (8) and (9) by a factor of 4/3,

$$\overline{\sigma}_{\alpha\beta\gamma}^{\mathcal{H}(1,2)} = \frac{4}{3}\overline{\sigma}_{\alpha\beta\gamma}^{1,2} = \frac{1}{3}\left(\sigma_{\alpha\beta\gamma} + \sigma_{\alpha\gamma\beta} - \sigma_{\beta\alpha\gamma} - \sigma_{\beta\gamma\alpha}\right), \tag{35}$$

$$\overline{\sigma}_{\alpha\beta\gamma}^{\mathcal{H}(1,3)} = \frac{4}{3}\overline{\sigma}_{\alpha\beta\gamma}^{1,3} = \frac{1}{3}\left(\sigma_{\alpha\beta\gamma} + \sigma_{\alpha\gamma\beta} - \sigma_{\gamma\beta\alpha} - \sigma_{\gamma\alpha\beta}\right). \tag{36}$$

Comparing with Eq. (24) we find

$$a_0^{+} = a_0^{-} = -a_1^{-} = -a_2^{+} = \frac{1}{3}, \quad a_1^{+} = a_2^{-} = 0 \tag{37}$$

in the case of Eq. (35), and

$$a_0^{+} = a_0^{-} = -a_1^{+} = -a_2^{-} = \frac{1}{3}, \quad , a_1^{-} = a_2^{+} = 0 \tag{38}$$

in the case of Eq. (36). Since both sets of coefficients satisfy the Hall-like conditions in Eq. (33), Eqs. (35) and (36) are generally valid expressions for the quadratic Hall conductivity.

## 4  Higher-order responses

At $n$-th order in the electric field, the Ohmic conductivity can be chosen as the fully symmetrized conductivity tensor obtained by setting $a_p = 1/(n+1)!$ for all $p$ in Eq. (19),

$$\sigma_{\alpha_0\alpha_1\ldots\alpha_n}^{\mathcal{O}} \equiv \hat{P}_{\mathcal{O}}\sigma_{\alpha_0\alpha_1\ldots\alpha_n} = \frac{1}{(n+1)!}\sum_p \sigma_{\alpha_{p(0)}\alpha_{p(1)}\ldots\alpha_{p(n)}}. \tag{39}$$

This generalizes to arbitrary $n$ the symmetrization procedure of Eqs. (3) and (14) for $n = 1$ and $n = 2$, respectively.

Let us now show that with the above choice of Ohmic projector, the Hall projector $\hat{P}_{\mathcal{H}} = \hat{1} - \hat{P}_{\mathcal{O}}$ satisfies Eqs. (22) and (23). We start again with the gauge invariance condition. Since the full $n$-th order current is by definition invariant under a gauge transformation $\Delta \sigma_{\alpha_0 \alpha_1 \ldots \alpha_n}$, the Hall part is invariant if and only if the Ohmic part is invariant. It is therefore sufficient to show that

$$\Delta \left( \hat{P}_{\mathcal{O}} j^{(n)}_{\alpha_0} \right) = \left( \hat{P}_{\mathcal{O}} \Delta \sigma_{\alpha_0 \alpha_1 \ldots \alpha_n} \right) E_{\alpha_1} \ldots E_{\alpha_n} \tag{40}$$

vanishes for arbitrary $E$. But since $\Delta \sigma_{\alpha_0 \alpha_1 \ldots \alpha_n}$ must vanish under symmetrization over the last $n$ indices to ensure that $\Delta j^{(n)} = 0$ (see Sec. 2), it also vanishes under full symmetrization by $\hat{P}_{\mathcal{O}}$. Next, it is clear that $\hat{P}_{\mathcal{O}}^2 \sigma_{\alpha_0 \alpha_1 \ldots \alpha_n} = \hat{P}_{\mathcal{O}} \sigma_{\alpha_0 \alpha_1 \ldots \alpha_n}$ because symmetrization of tensor that is already fully symmetric does not change it further. Therefore,

$$\hat{P}_{\mathcal{H}}^2 j^{(n)} = \left( 1 - 2\hat{P}_{\mathcal{O}} + \hat{P}_{\mathcal{O}}^2 \right) j^{(n)} = \left( 1 - \hat{P}_{\mathcal{O}} \right) j^{(n)} = \hat{P}_{\mathcal{H}} j^{(n)} . \tag{41}$$

Finally, from the $n$-th order generalization of Eq. (15) it follows that $j^{(n)}_{\mathcal{H}} = j^{(n)} - j^{(n)}_{\mathcal{O}}$ is dissipationless. Thus we have obtained a solution that satisfies Eqs. (22) and (23) at any order in $E$, and found that it corresponds to the Hall vs Ohmic partition of the current.

To recapitulate, one can always define the Ohmic part of the $n$-th order conductivity as the totally symmetric part, and the Hall part as the remainder. For $n = 1$, this procedure reduces to the standard partition of the linear conductivity according to Eqs. (2) and (3). We demonstrated in Sec. 3 that for $n = 2$ the same procedure leads to the only well-defined (idempotent and gauge-invariant) generic partition of the quadratic current, and in Appendix A we generalize that proof to arbitrary $n$.

## 5   Symmetry analysis of the quadratic dc conductivity

At linear order in $E$, the Hall vs Ohmic decomposition is intimately related to time-reversal symmetry $\mathcal{T}$ by virtue of the Onsager reciprocity relation

$$\sigma_{\alpha\beta}(H, M) = \sigma_{\beta\alpha}(-H, -M) . \tag{42}$$

It follows from this relation that the Ohmic part of the linear response is $\mathcal{T}$-even, while the Hall part is $\mathcal{T}$-odd [27, 28]. In the nonlinear regime, both Hall and Ohmic responses can have $\mathcal{T}$-even and $\mathcal{T}$-odd components; this gives four contributions in total, of which only three are present in Eq. (10) for the disorder-free $\sigma_{\alpha\beta\gamma}$. The reason why there is no $\mathcal{T}$-even Ohmic term in Eq. (10) is that in the semiclassical wavepacket formalism there is no correction to the band energy at first order in the electric field [29]; the leading correction is of second order, and it contributes to the $\mathcal{T}$-even cubic conductivity [24].

We will see shortly that $\sigma_{\alpha\beta\gamma}$ is purely Ohmic and $\mathcal{T}$-even in five of the 122 magnetic point groups. Since for materials in those point groups the disorder-free part of $\sigma_{\alpha\beta\gamma}$ vanishes identically, their symmetry-allowed quadratic response must be entirely disorder-mediated; this is consistent with the finding that a skew-scattering contribution to $\sigma_{\alpha\beta\gamma}$ is present in all non-centrosymmetric materials [30]. Contributions from disorder to $\sigma^{\mathcal{H}}_{\alpha\beta\gamma}$ have been studied recently [15, 31, 32], but similar contributions to $\sigma^{\mathcal{O}}_{\alpha\beta\gamma}$ have received little attention so far. In this regard, we note that the expressions for $\sigma_{\alpha\beta\gamma}$ obtained in Refs. [15, 30–32] contain not only Hall-like but also Ohmic parts, which can be separated out using Eqs. (14) and (16).

Table 1: Decomposition of the quadratic conductivity into Ohmic vs Hall parts and $\mathcal{T}$-even vs $\mathcal{T}$-odd parts. The Ohmic part is represented by a totally symmetric rank-3 polar tensor [Eq. (14)], and the Hall part by a traceless rank-2 axial tensor [Eq. (44)]. Each entry in the table denotes the corresponding Jahn symbol [33].

|  | Quadratic Ohmic | Quadratic Hall |
|---|---|---|
| $\mathcal{T}$-even | $[V^3]$ | $eV^2$ (traceless part) |
| $\mathcal{T}$-odd | $a[V]^3$ | $aeV^2$ (traceless part) |

In preparation for performing a symmetry analysis of $\sigma_{\alpha\beta\gamma}$, let us count the number of independent coefficients needed to describe the quadratic Ohmic and Hall responses in two-dimensional (2D) and three-dimensional (3D) space. As $\sigma_{\alpha\beta\gamma}$ can be chosen to be symmetric in the last two indices, it has 6 (18) independent components in 2D (3D). $\sigma^{\mathcal{O}}_{\alpha\beta\gamma}$ can be chosen to be fully symmetric, and hence it has 4 (10) independent components in 2D (3D), leaving $\sigma^{\mathcal{H}}_{\alpha\beta\gamma}$ with $6 - 4 = 2$ ($18 - 10 = 8$) independent components in 2D (3D). Those Hall-like components can be repackaged as an axial vector in 2D, and as a traceless[2] rank-2 axial tensor in 3D. Choosing the latter as

$$\chi^{\mathcal{H}}_{\gamma\delta} = \frac{1}{2}\varepsilon_{\alpha\beta\gamma}\overline{\sigma}^{\mathcal{H}(1,2)}_{\alpha\beta\delta} = \frac{1}{2}\varepsilon_{\alpha\beta\gamma}\overline{\sigma}^{\mathcal{H}(1,3)}_{\alpha\delta\beta}, \tag{43}$$

and using either Eq. (35) or Eq. (36), one finds

$$\chi^{\mathcal{H}}_{\gamma\delta} = \frac{1}{3}\varepsilon_{\alpha\beta\gamma}\left(\sigma_{\alpha\beta\delta} + \sigma_{\alpha\delta\beta}\right). \tag{44}$$

The tensor $\chi^{\mathcal{H}}$ remains invariant under gauge transformations of the quadratic conductivity [Eq. (5)]. This gauge-invariant repackaging of the quadratic Hall conductivity tensor is analogous to the repackaging $\varepsilon_{\gamma\alpha\beta}\sigma_{\alpha\beta}/2$ of the linear Hall conductivity as an axial vector. As an example, in Appendix B we evaluate $\chi^{\mathcal{H}}$ for the disorder-free quadratic conductivity (10).

According to the preceeding analysis, the quadratic conductivity can be divided quite generally into an Ohmic part given by a totally symmetric rank-3 polar tensor [Eq. (14)], and a Hall part expressible as a traceless rank-2 axial tensor [Eq. (44)]. Each of these can be further decomposed into $\mathcal{T}$-even and $\mathcal{T}$-odd parts, resulting in a total of four contributions whose Jahn symbols [33] are indicated in Table 1.

Taking the Jahn symbols in Table 1 as input, we have used the MTENSOR program [34] hosted on the the Bilbao Crystallographic Server (http://www.cryst.ehu.es/cryst/mtensor) to obtain the symmetry-adapted forms of the four contributions to the quadratic conductivity in each magnetic point group. The results are summarized in Table 2, where we indicate the existence or absence of each contribution in each point group. The rows of the table are organized into four blocks: in the first block the quadratic response is entirely absent, in the second (third) it is purely Hall-like (Ohmic), and in the fourth both Hall and Ohmic responses are present. Since we have not invoked specific microscopic mechanisms in setting up Table 2, our symmetry analysis is purely phenomenological. (If the last column is removed and the table is rearranged accordingly, it reduces to the table given in Ref. [22], which pertains to the three terms in Eq. (10) for the disorder-free $\sigma_{\alpha\beta\gamma}$.) Interestingly, all $2^4 = 16$ possibilities are realized in Table 2. In particular, there are magnetic point groups for which only one of the four

---

[2]The fact that $\chi^{\mathcal{H}}$ is traceless went unnoticed in Ref. [15], where it is stated that $\chi^{\mathcal{H}}$ has nine independent components rather than eight.

Table 2: Magnetic point groups classified by the existence or absence of the four symmetry types of quadratic conductivities in a vanishing external magnetic field.

| Magnetic point groups | Quadratic Hall | | Quadratic Ohmic | |
|---|---|---|---|---|
| | $\mathcal{T}$-odd | $\mathcal{T}$-even | $\mathcal{T}$-odd | $\mathcal{T}$-even |
| $-1$, $-11'$, $2/m$, $2/m1'$, $2'/m'$, mmm, mmm1', m'm'm, $4/m$, $4/m1'$, $4'/m$, $4/mmm$, $4/mmm1'$, $4'/mm'm$, $4/mm'm'$, $-3$, $-31'$, $-3m$, $-3m1'$, $-3m'$, $6/m$, $6/m1'$, $6'/m'$, $6/mmm$, $6/mmm1'$, $6'/m'mm$, $6/mm'm'$, m$-3$, m$-31'$, 432, 4321', m$-3$m, m$-3$m1', m$-3$m', $m'-3'm$ | ✗ | ✗ | ✗ | ✗ |
| $4/m'm'm'$, $6/m'm'm'$ | ✓ | ✗ | ✗ | ✗ |
| 4221', 6221' | ✗ | ✓ | ✗ | ✗ |
| 422, 622 | ✓ | ✓ | ✗ | ✗ |
| $6'/m$, $6'/mmm'$, $m'-3$, $4'32'$, $m'-3m$ | ✗ | ✗ | ✓ | ✗ |
| $-61'$, $-6m21'$, 231', $-43m1'$, $-4'3m'$ | ✗ | ✗ | ✗ | ✓ |
| $-6$, $-6m2$, $-6m'2'$, 23, $-43m$ | ✗ | ✗ | ✓ | ✓ |
| $-6'm'2$ | ✓ | ✗ | ✗ | ✓ |
| $6'22'$ | ✗ | ✓ | ✓ | ✗ |
| $-1'$, $2'/m$, $2/m'$, m'mm, m'm'm', $4/m'$, $4'/m'$, $4/m'mm$, $4'/m'm'm$, $-3'$, $-3'm$, $-3'm'$, $6/m'$, $6/m'mm$ | ✓ | ✗ | ✓ | ✗ |
| 11', 21', m1', 2221', mm21', 41', $-41'$, 4mm1', $-42m1'$, 31', 321', 3m1', 61', 6mm1' | ✗ | ✓ | ✗ | ✓ |
| $4'22'$, $42'2'$, $62'2'$ | ✓ | ✓ | ✓ | ✗ |
| 4m'm', $-4'2m'$, 6m'm' | ✓ | ✓ | ✗ | ✓ |
| $-6'$, $-6'm2'$ | ✓ | ✗ | ✓ | ✓ |
| $6'$, $6'mm'$ | ✗ | ✓ | ✓ | ✓ |
| 1, 2, 2', m, m', 222, 2'2'2, mm2, m'm2', m'm'2, 4, 4', $-4$, $-4'$, 4mm, 4'm'm, $-42m$, $-4'2'm$, $-42'm'$, 3, 32, 32', 3m, 3m', 6, 6mm | ✓ | ✓ | ✓ | ✓ |

contributions is present; clearly, materials belonging to those point groups should be ideally suited for studying one specific type of quadratic current response. As already mentioned, in the point groups where that response is purely Ohmic and $\mathcal{T}$-even the quadratic current is purely disorder mediated.

## 6 Discussion

In this work we have shown how, given a dc conductivity tensor of arbitrary order $n$ in the electric field, the current may be uniquely separated into Hall and Ohmic parts,

$$\boldsymbol{j}^{(n)} = \boldsymbol{j}_{\mathcal{H}}^{(n)} + \boldsymbol{j}_{\mathcal{O}}^{(n)}, \tag{45}$$

by taking linear combinations of that tensor with permuted indices. This separation is insensitive to the particular gauge choice for the conductivity, and applying it multiple times gives the same result as applying it only once. No other generic order-by-order partition of the induced current fulfills these two requirements. Thus, once we have separated the Hall and Ohmic parts we cannot make any further subdivisions of the current into physically meaningful parts

without invoking either the symmetries of the system, or the microscopic processes producing the nonlinear currents.

The nonlinear Hall effect has sometimes been associated with the transverse part of the current [24], and spontaneous unidirectional magnetoresistence with a longitudinal response [8, 9]. The present work provides sharper definitions of Hall and Ohmic nonlinear responses that are generally valid irrespective of crystal symmetry. For example, spontaneous unidirectional magnetoresistence should be defined as the $\mathcal{T}$-odd part of the quadratic Ohmic response, which generally has both longitudinal and transverse components. This is consistent with the analysis in Ref. [10], where the same conclusion was reached on the basis of a particular mechanism, namely the nonlinear Drude term in Eq. (10). We hope that the present work will be useful for identifying the Hall and Ohmic parts of nonlinear responses, both experimentally and in the context microscopic theories.

## Acknowledgements

We thank David Vanderbilt for comments on the manuscript, and Cheol-Hwan Park and José Luís Martins for discussions.

### Funding information

Work by S.S.T. was supported by the European Research Council (ERC) under the European Union's Horizon 2020 research and innovation program (ERC-StG-Neupert757867-PARATOP), and by Grant No. PP00P2-176877 from the Swiss National Science Foundation. Work by I.S. was supported by Grant No. FIS2016-77188-P from the Spanish Ministerio de Economía y Competitividad.

## A  Uniqueness of the partition at arbitrary order

In this Appendix we prove that the Hall vs Ohmic partition of the current described in Sec. 4 is the only valid generic partition at arbitrary order $n$ in the electric field. We start with the general expression in Eq. (19) for the action of the operator $\hat{P}$ on the conductivity,

$$\hat{P}\sigma_{\alpha_0\alpha_1\ldots\alpha_n} = \sum_p a_p \sigma_{\alpha_{p(0)}\alpha_{p(1)}\ldots\alpha_{p(n)}}, \tag{46}$$

where the sum is over all permutations $\{p(1),\ldots,p(n)\}$ of $\{0,1,\ldots,n\}$. The generalization of Eq. (25) for the action of $\hat{P}$ on the current reads

$$\begin{aligned}
\hat{P}j^{(n)}_{\alpha_0} = \big(& A_0\sigma_{\alpha_0\alpha_1\ldots\alpha_n} + A_1\sigma_{\alpha_1\alpha_0\ldots\alpha_n} + \ldots \\
& + A_i\sigma_{\alpha_1\ldots\alpha_i\alpha_0\alpha_{i+1}\ldots\alpha_n} + \ldots + A_n\sigma_{\alpha_1\ldots\alpha_n\alpha_0}\big)E_{\alpha_1}\ldots E_{\alpha_n},
\end{aligned} \tag{47}$$

where

$$A_i = \sum_p^{\overset{p(i)=0}{\frown}} a_p. \tag{48}$$

Since they fully determine the projected current, the $A_i$ are the only physically meaningful parameters, and changes in the parameters $a_p$ that leave every $A_i$ invariant amount to gauge transformations.

Recall from Sec. 2 that a general gauge transformation $\Delta\sigma_{\alpha_0\ldots\alpha_n}$ of the conductivity tensor must satisfy the condition

$$\sum_q \Delta\sigma_{\alpha_0\alpha_{q(1)}\ldots\alpha_{q(n)}} = 0, \tag{49}$$

where the summation is over all permutations $\{q(1),\ldots,q(n)\}$ of $\{1,\ldots,n\}$. As stated in Eq. (23), we want the projected current to be invariant under all possible gauge transformations. To make progress, it is sufficient to require at this point invariance under the subset of gauge transformations $\Delta\sigma^i_{\alpha_0\alpha_1\ldots\alpha_n}$ that are antisymmetric under permutation of the indices at positions $i$ and $i+1$,

$$\Delta\sigma^i_{\alpha_0\ldots\alpha_{i-1}\alpha_i\alpha_{i+1}\alpha_{i+2}\ldots\alpha_n} = -\Delta\sigma^i_{\alpha_0\ldots\alpha_{i-1}\alpha_{i+1}\alpha_i\alpha_{i+2}\ldots\alpha_n}, \tag{50}$$

where $0 < i < n$. For such transformations, the gauge invariance condition on the projected current (47) takes the form

$$\left(A_i - A_{i+1}\right)\Delta\sigma^i_{\alpha_1\ldots\alpha_i\alpha_0\alpha_{i+1}\alpha_{i+2}\ldots\alpha_n}E_{\alpha_1}\ldots E_{\alpha_n} = 0. \tag{51}$$

This condition can hold in general if and only if $A_i = A_{i+1}$, and by letting the index $i$ run from 1 to $n-1$ we get

$$A_1 = A_2 = \ldots = A_n. \tag{52}$$

Therefore, the two parameters $A_0$ and $A_1$ fully determine the projected current.

Let us turn now to the idempotency condition (22). Acting with $\hat{P}$ on both sides of Eq. (46) we obtain the following generalization of Eq. (27),

$$\hat{P}^2\sigma_{\alpha_0\alpha_1\ldots\alpha_n} = \sum_p \tilde{a}_p\sigma_{\alpha_{p(0)}\alpha_{p(1)}\ldots\alpha_{p(n)}} = \sum_{p,p_1,p_2}^{p_2\cdot p_1 = p} a_{p_1}a_{p_2}\sigma_{\alpha_{p(0)}\alpha_{p(1)}\ldots\alpha_{p(n)}}, \tag{53}$$

and hence the idempotency condition becomes $A_i = \tilde{A}_i$ for $i = 0,\ldots,n$ where, by analogy with Eq. (48),

$$\tilde{A}_i \equiv \sum_p^{p(i)=0} \tilde{a}_i = \sum_p^{p(i)=0}\sum_{p_1,p_2}^{p_2\cdot p_1 = p} a_{p_1}a_{p_2}. \tag{54}$$

Solving this equation for arbitrary $n$ is not as easy as solving it for $n = 2$ [Eq. (30)]. But having settled the gauge invariance conditions in Eq. (52), we can now pick a convenient gauge for the coefficients $a_p$. (This entails no loss of generality, because we study the action of $\hat{P}$ on the physical current, not on a particular form of the conductivity tensor.) We choose the most symmetric gauge compatible with Eq. (52), namely, the gauge where all terms in the summand of Eq. (48) are identical,

$$a_p = \begin{cases} A_0/n!, & \text{if } p(0) = 0 \\ A_1/n!, & \text{if } p(0) \neq 0 \end{cases}. \tag{55}$$

Substituting in Eq. (54), the idempotency condition $A_i = \tilde{A}_i$ becomes

$$A_i = \frac{1}{n!^2}\sum_p^{p(i)=0}\left(\sum_{p_1,p_2}^{\substack{p_2\cdot p_1 = p \\ p_1(0)=0 \\ p_2(0)=0}}A_0^2 + \sum_{p_1,p_2}^{\substack{p_2\cdot p_1 = p \\ p_1(0)=0 \\ p_2(0)\neq 0}}A_0 A_1 + \sum_{p_1,p_2}^{\substack{p_2\cdot p_1 = p \\ p_1(0)\neq 0 \\ p_2(0)=0}}A_1 A_0 + \sum_{p_1,p_2}^{\substack{p_2\cdot p_1 = p \\ p_1(0)\neq 0 \\ p_2(0)\neq 0}}A_1 A_1\right), \tag{56}$$

with $i$ running from 0 to $n$. Due to Eq. (52), the $n$ equations with $1 \leq i \leq n$ are identical, leaving two equations only. These can be written as

$$A_0 = \frac{1}{n!^2}\left(aA_0^2 + bA_0 A_1 + cA_1^2\right), \qquad A_1 = \frac{1}{n!^2}\left(dA_0^2 + eA_0 A_1 + fA_1^2\right), \tag{57}$$

where the coefficients $a$ to $f$ are the numbers of pairs of permutations $p_1, p_2$ of the set $\{0, 1, \ldots n\}$ that satisfy the conditions

$$
a, d \quad : \quad p_1(0) = p_2(0) = 0, \tag{58a}
$$

$$
b, e \quad : \quad (p_1(0) = 0 \wedge p_2(0) \neq 0) \vee (p_1(0) \neq 0 \wedge p_2(0) = 0), \tag{58b}
$$

$$
c, f \quad : \quad p_1(0) \neq 0 \wedge p_2(0) \neq 0, \tag{58c}
$$

together with

$$
a, b, c \quad : \quad p_2(p_1(0)) = 0, \tag{58d}
$$

$$
d, e, f \quad : \quad p_2(p_1(1)) = 0. \tag{58e}
$$

It now becomes a straightforward combinatorial exercise to obtain

$$
\begin{aligned}
a &= n!^2, & d &= 0, \\
b &= 0, & e &= 2 \cdot n!^2, \\
c &= n \cdot n!^2, & f &= (n-1) \cdot n!^2,
\end{aligned} \tag{59}
$$

which leads to the following generalization of Eq. (31),

$$
A_0 = A_0^2 + n A_1^2, \qquad A_1 = 2 A_0 A_1 + (n-1) A_1^2. \tag{60}
$$

Apart from the trivial solutions $\hat{P}_0$ and $\hat{P}_1$ of the same type as in Eq. (32), these equations have the solutions

$$
\begin{cases}
\hat{P}_{\mathcal{H}} : (A_0, A_1 = \ldots = A_n) = \left( \frac{n}{n+1}, -\frac{1}{n+1} \right), \\
\hat{P}_{\mathcal{O}} : (A_0, A_1 = \ldots = A_n) = \left( \frac{1}{n+1}, \frac{1}{n+1} \right),
\end{cases} \tag{61}
$$

which generalize Eq. (33). It can be readily verified that the solution for $\hat{P}_{\mathcal{O}}$ is satisfied by Eq. (39). And since $\hat{P}_{\mathcal{H}}$ fulfills the Hall condition (34) but $\hat{P}_{\mathcal{O}}$ does not, we have obtained a unique partition of the $n$-th order current into Hall and Ohmic components.

## B  Repackaging of the disorder-free quadratic Hall conductivity

Inserting Eq. (10) for $\sigma_{\alpha\beta\gamma}$ in Eq. (44) for $\chi_{\gamma\delta}^{\mathcal{H}}$ and writing $\Omega_n^{\alpha\beta} = \varepsilon_{\alpha\beta\gamma} \Omega_n^{\gamma}$ one finds

$$
\chi_{\alpha\beta}^{\mathcal{H}} = \frac{e^3}{\hbar} \int_{kn} f_0(\epsilon_n) \varepsilon_{\alpha\gamma\delta} \partial_\gamma G_n^{\delta\beta} + \frac{e^3 \tau}{\hbar^2} \left[ D_{\beta\alpha} - \frac{1}{3} \delta_{\alpha\beta} \mathrm{Tr}(D) \right], \tag{62}
$$

where

$$
D_{\beta\alpha} = \int_{kn} f_0(\epsilon_n) \partial_\beta \Omega_n^{\alpha} \tag{63}
$$

is the Berry curvature dipole [14]. The first in Eq. (62) agrees with the expression obtained in Ref. [23] starting from the gauge-dependent definition $\chi_{\gamma\delta}^{\mathcal{H}} = \varepsilon_{\alpha\beta\gamma} \sigma_{\alpha\beta\delta}/2$. The second term agrees with the expression in Eq. (8) of Ref. [15], once that expression is multiplied by the factor of 4/3 that was discussed in connection with Eqs. (35) and (36). That second term can be simplified by noting that $\mathrm{Tr}(D) = 0$ for topological reasons [31, 35], yielding

$$
\chi_{\alpha\beta}^{\mathcal{H}} = \frac{e^3}{\hbar} \int_{kn} f_0(\epsilon_n) \left[ \varepsilon_{\alpha\gamma\delta} \partial_\gamma G_n^{\delta\beta} + (\tau/\hbar) \partial_\beta \Omega_n^{\alpha} \right] \tag{64}
$$

for the disorder-free quadratic Hall tensor. The first term is $\mathcal{T}$ odd and intrinsic (independent of $\tau$), and the second is $\mathcal{T}$ even and extrinsic.

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
