# Peer review of "On the separation of Hall and Ohmic nonlinear responses"

_SciPost Physics, doi:SciPost Phys. Core 5, 039 (2022)_

## Round 1 · Referee Report · Anonymous · 2022-5-26

Report

After reading the comments to my previous report and the changes made in the manuscript, I consider the manuscript acceptable for publication in SciPost Physics.

The symmetry analysis added to Sec. 5 might be useful for experimentalists to properly identify ohmic vs. Hall contributions, in special for magnetic groups, which are now in the spot of searching time reversal breaking semimetals. Determining the proper decomposition of Hall responses (linear and nonlinear) in such systems appears to be of paramount importance to compare experiments with theoretical results.

---

## Round 1 · Author Response

We thank the Referee for a thoughtful report. Below we provide replies to the questions and concerns raised in the report, and describe the changes that have been made to the manuscript to address those concerns.

In the "Weaknesses" section, the Referee writes:

2- The manuscript mainly focuses on the generic definition of currents with respect to the permutation of electric fields. It is difficult (to me) to foresee practical consequences of the present manuscript in real experiments.

As a minor clarification, we note that the permutations we consider are not just with respect to the electric fields (represented by the last n indices of the n-th order conductivity tensor), but with respect to all indices, including the 0-th index representing the current: see Eqs .(19,20).

As for the practical consequences of our analysis, they are discussed below.

In the third paragraph of the Report, the Referee writes:

But it is not clear from the manuscript if different prescriptions for nonlinear conductivities have any experimental impact, beyond precise values of parameters. If I understand it well, the authors discuss that different prescriptions lead to different overall numerical prefactors, that can be accounted for in unknown parameters, like lifetimes or similar.

In the specific example discussed around Eqs.(6-13), different prescriptions that do not pass our sanity criteria give different overall numerical prefactors for the Hall current. More generally, incorrect Hall vs Ohmic partitions may also lead to Hall currents point along different directions. Moreover, Eq.(17) shows that even when the Hall current only changes by a numerical prefactor, the Ohmic current -- which in general is not parallel to the applied field -- may change direction. Finally, the numerical prefactors are important by themselves to allow the experimental benchmarking of parameter-free theories. In particular, the lifetimes may either be evaluated from first principles, or extracted from the measured linear Ohmic resistivity.

In the fourth paragraph of the Report, the Referee writes:

Taking the second order conductivity as an example, it is well known that inversion symmetry must be broken to get a non-zero response. According to the authors, this information is irrelevant for the partitioning they describe, but is of paramount importance in experiments, so I find this shift between what is important in the present manuscript, a little bit confusing.

As the Referee points out, if inversion symmetry is present the entire quadratic conductivity tensor vanishes; this is now mentioned explicitly below Eq. 4. In that case, our prescription gives the trivially correct result that the Hall and Ohmic parts of the quadratic current response vanish separately. Our analysis becomes nontrivial when the quadratic current is symmetry allowed, i.e., in acentric crystals, as elaborated in our answer to the comment below.

In the fifth and final paragraph of the Report and in the Requested Changes, the Referee writes:

To frame a little bit my concern, the question I would find quite useful to be answered in the present manuscript is How the present discussion about partitioning might be relevant to understand experimental results? Discussing it for the second nonlinear conductivity would be enough (for me).

1- I strongly encourage the authors to enlarge the discussion section with the role of inversion symmetry and experimental consequences of their results in the case of second nonlinear conductivity, paying attention to any available physical example (the way shift , photogalvanic currents modify when performing the partitioning procedure, etc).

To address this concern and following the subsequent recommendation, we have made substantial revisions to the manuscript, namely:

  • We added a new Sec. 5 where we carry out a systematic symmetry analysis of the quadratic conductivity, broken down into four fundamental contributions: Hall vs Ohmic, and time-reversal even vs time-reversal odd. The results, summarized in Table 2, could guide the search for materials displaying only one type of quadratic response. We call attention to the time-even Ohmic component, which is purely disorder mediated. We correct a result from the recent literature, and point out a possible oversight in recent papers on disorder-mediated quadratic responses, where the Ohmic part of those response may have been overlooked. These changes are reflected in part in the expanded abstract.

  • We revised the Discussion section, which now contains two paragraphs only. The second paragraph is new. In it, we comment on the relevance of our analysis for interpreting recent measurements of (i) a higher-order (cubic) nonlinear current, and (ii) spontaneous unidirectional magnetoresistance. In particular, we provide a sharp phenomenological definition of the latter effect, without invoking specific microscopic mechanisms.

---

## Round 1 · List of Changes

* We have reverted to the title of the original submission.

* A new section (Sec. 5) was added, and the abstract eas extended to
reflect this change.

* The Discussion section (Sec. 6) was revised.

* A new Appendix B was added.

* Some changes were made to the bibliography.

* Miscellaneous edits were made across the manuscript to improve
clarity.

---

## Editorial Decision

published